# Cluster-Masked Scanning and Pretraining for Enhanced xLSTM Vision Performance

## Abstract

While modern recurrent architectures like xLSTM show promise for vision tasks, their potential has been hindered by the challenge of effectively applying autoregressive pretraining—a cornerstone of NLP success—to 2D image data. This paper introduces MAL, a framework that unlocks autoregressive learning for vision-oriented xLSTMs. Our core innovation is a **cluster-masked pretraining strategy**, which reorganizes an image into a sequence of semantically meaningful local clusters. This approach creates a more structured input sequence uniquely suited to xLSTM's memory mechanisms. By combining this with our **novel cluster scanning strategy** which defines an optimal processing order, MAL effectively learns powerful visual representations by predicting entire image regions autoregressively. Our experiments show that this novel pretraining scheme allows MAL to significantly outperform traditional supervised models, fully leveraging the scaling potential of xLSTM and setting a new performance benchmark.

## 1 Introduction

In recent years, efficient visual representation learning has become a key focus in computer vision research. The introduction of Transformer models and State Space Models (SSM), like Mamba, has significantly impacted visual task processing, showing impressive performance across various applications Liu et al. (2024b); Ma et al. (2024). However, these models often face challenges when scaling to larger sizes, which limits their efficiency and applicability Hatamizadeh & Kautz (2024). For instance, Vision Mamba (Vim) Zhu et al. (2024) can experience performance stagnation or training crashes at larger scales Ren et al. (2024).

In this paper, we focus on autoregressive pretraining in self-supervised visual representation learning, which predicts the next token sequentially from start to finish. This approach is motivated by two key factors. Firstly, autoregressive pretraining is a standard method for training large language models and has been influential across various architectures, including Transformers and Mamba Gu & Dao (2023); Liu & Yi (2025). It has shown promise in computer vision, as evidenced by Vision Transformer (ViT) El-Nouby et al. (2024); Ren et al. (2023a). Secondly, the Mamba architecture's linear attention properties naturally support autoregressive modeling by allowing each token to attend only to its predecessors, enhancing training efficiency.

The development of the extended Long Short-Term Memory (xLSTM) family marks a significant advancement in natural language processing (NLP). xLSTM enhances traditional LSTM architecture, achieving performance comparable to leading Transformer models while overcoming some LSTM limitations. The vision-LSTM approach has successfully adapted xLSTM for visual tasks, demonstrating its versatility. Inspired by these advancements, we propose a novel approach that utilizes xLSTM instead of Mamba or Transformer components to construct a visual autoregressive pretraining framework.

To address these challenges and unlock the full potential of xLSTM in vision, we propose MAL (Cluster-Masked Scanning and Pretraining for Enhanced xLSTM Vision Performance). Instead of treating an image as a simple flat sequence of patches, which can disrupt local spatial coherence, MAL introduces a novel **cluster-masked autoregressive pretraining** methodology. Our key insight is that by grouping spatially adjacent patches into larger, semantically richer clusters, we can create a more structured sequence for the model to process. This approach not only enhances local feature capture but also presents a more manageable and meaningful prediction task for the autoregressive

objective. Furthermore, MAL integrates this strategy within a parallel encoder-decoder architecture, demonstrating for the first time an effective way to apply autoregressive pretraining to the xLSTM family for visual representation learning.

Experimental results show that MAL significantly outperforms traditional supervised training models, effectively leveraging xLSTM's scaling potential to handle large and complex visual datasets. By addressing LSTM limitations with xLSTM's advanced capabilities and novel pretraining strategies, MAL sets a new standard for visual task performance, highlighting the transformative potential of autoregressive in computer vision. The major contributions of this paper are three-fold:

- **Innovative Cluster-Masked Masking Strategy**: The paper introduces a novel cluster-masked masking approach that enhances xLSTM's ability to capture local image features and optimizes image scanning efficiency. This method groups spatially adjacent patches into larger clusters, improving both feature extraction and computational efficiency.

- **Cluster-Masked Scanning Method**: Our framework introduces a new scanning strategy that processes these clustered representations, allowing for more effective autoregressive modeling. This method enhances the model's understanding of spatial relationships within the visual data.

- **Improved Model Adaptability and Performance**: Our approach reduces discrepancies by maintaining architectural consistency between pretraining and fine-tuning. To our knowledge, this is the first use of xLSTM for autoregressive tasks in visual representation learning. Experiments show it significantly outperforms traditional supervised models across various visual tasks, effectively leveraging xLSTM's scalability.

## 2 RELATED WORK

### 2.1 LSTM IN VISION

Recurrent Neural Networks (RNNs) were initially developed to address problems in Natural Language Processing (NLP), such as time-series prediction and speech recognition, by effectively capturing temporal dependencies in sequential data. Recently, to overcome the quadratic computational complexity of transformers, time-parallel data-dependent RNNs (referred to as linear RNNs in this paper) have made significant advancements Qin et al. (2023); Orvieto et al. (2023); Sun et al. (2023); De et al. (2024); Yang et al. (2023); Gu & Dao (2023); Sun et al. (2024); Beck et al. (2024). ViG's Liao et al. (2024) innovative use of gated linear attention to achieve linear complexity by dynamically adjusting receptive fields. These models provide efficient parallel training capabilities while maintaining linear complexity, achieving performance levels that meet or even exceed those of transformers. Due to their scalability and efficiency, linear RNNs are expected to play an increasingly important role across various fields, with some studies Duan et al. (2024a); Alkin et al. (2024a); Liang et al. (2024) already applying linear RNNs to the 2D vision domain. Vision-LSTM (ViL), which adapts the xLSTM building blocks for computer vision, has been shown to outperform the ViT training pipeline—a result of years of hyperparameter tuning and transformer improvements. This paper aims to extend linear RNNs to 2D self-supervised visual representation tasks, thanks to their ability to model long-range dependencies.

### 2.2 SELF-SUPERVISED VISUAL REPRESENTATION LEARNING

Self-supervised visual representation learning seeks to develop robust, transferable representations without labelled data, using methods like contrastive learning Chen et al. (2020c); He et al. (2020); Chen et al. (2021; 2020b), position prediction Zhai et al. (2022), and masked image modeling He et al. (2022); Bao et al. (2022); Ren et al. (2023b). This paper focuses on autoregressive pretraining, a successful NLP technique that has been explored less in computer vision. iGPT Chen et al. (2020a) first introduced generative pretraining transformers to vision, showcasing the potential of autoregressive pretraining in self-supervised learning. Enhancements by SAIM Qi et al. (2023) and RandSAC Hua et al. (2022) used the ViT architecture and random sequence permutation, achieving results comparable to MAE He et al. (2022). D-iGPT Ren et al. (2023a) adjusted the learning objective to predict both the next and visible tokens. AIM El-Nouby et al. (2024) demonstrated that ViT could scale effectively with increased model capacity and data volume. Unlike these studies focused

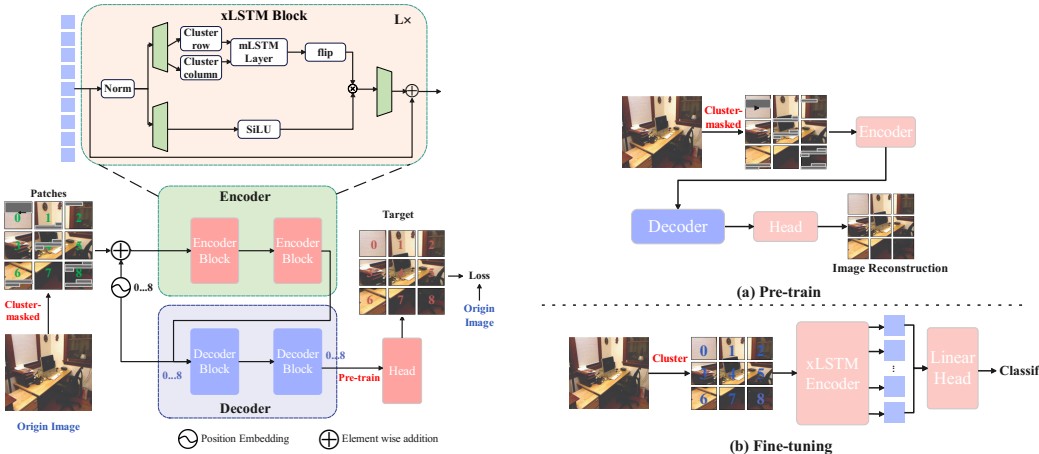

Figure 1: Overall architecture.    Figure 2: Pretrain and fine-tuning.

on transformers, our work is the first to explore autoregressive visual pretraining with the xLSTM architecture.

## 3 METHODS

We introduce the MAL framework, which enhances autoregressive visual representation learning by leveraging xLSTM with a cluster-masked scanning strategy and an encoder-decoder pretraining approach. The framework transitions from pixel-based to patch-based prediction units and explores different cluster prediction orders.

### 3.1 VISION LSTM ENCODER

As depicted in Figure 1, the MAL encoder is built with alternating mLSTM blocks that are fully parallelizable, featuring a matrix memory with a covariance update rule. Each mLSTM block incorporates an input gate, a forget gate, and multi-head layer normalization, all parameterized with linear layers. This design enables the Vision LSTM Encoder to effectively capture dependencies across the image, enhancing its ability to model complex visual patterns. Notably, the mLSTM serves as a component of the xLSTM Beck et al. (2024) framework introduced in this work. **With further xLSTM details provided in the appendix Section xLSTM Block.**

### 3.2 AUTOREGRESSIVE PRETRAINING

First, we briefly revisit autoregressive pretraining in NLP. Then, we focus on autoregressive pretraining with xLSTM in vision, including the prediction unit and prediction order design.

#### 3.2.1 EVOLUTION OF PREDICTION UNITS

**Pixel-based Prediction Unit.**    Transitioning from 1D sentences to 2D images requires defining an appropriate autoregressive prediction unit. Initially, as in iGPT Chen et al. (2020a), each pixel serves as the prediction unit (see Fig. 3(b)). For an image $X = \{p_1, ..., p_n\}$, our objective is to minimize the loss function:

$$\mathcal{L} = \sum_{i=1}^{n-1} l(f([p_1, \ldots, p_i]), p_{i+1}), \quad l(\hat{y}, y) = |\hat{y} - y|^2. \tag{1}$$

Here $f(\cdot)$ denotes the xLSTM model, and $p_i$ represents the image's $i_{th}$ pixel.

**Patch-based Prediction Unit.**    We can use a patch-based method to address the computational challenges of pixel-based approaches in high-resolution images, as highlighted in the iGPT pa-

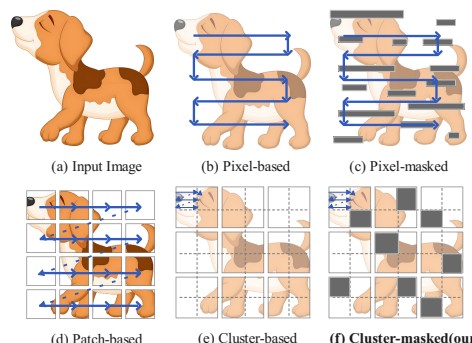

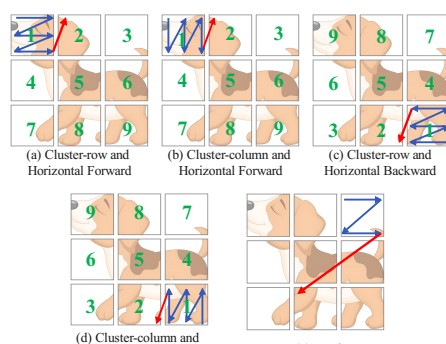

Figure 3: Different prediction units in the autoregressive modeling.

Figure 4: Different cluster prediction orderings of a visual sentence.

per Chen et al. (2020a). We effectively reduce the sequence length by dividing images into non-overlapping patches, similar to the method in Dosovitskiy et al. (2020). For instance, an image of size 224×224 can be transformed from a sequence of 50,176 pixels (as in iGPT) to just 196 patches using a 16 × 16 patch size, where $P_i \in \mathcal{R}^{16 \times 16}$ is the $i_{th}$ patch. This shift from predicting pixels Chen et al. (2020a) to predicting patches Dosovitskiy et al. (2020); Zhu et al. (2024); El-Nouby et al. (2024), as illustrated in Figure 3(d), reformulates the autoregressive input to $X = \{P_1, ..., P_n\}$:

$$\mathcal{L} = \sum_{i=1}^{n-1} l(f([P_1, \dots, P_i]), P_{i+1}), \quad l(\hat{y}, y) = |\hat{y} - y|^2. \tag{2}$$

### 3.2.2 CLUSTER-MASKED ENHANCED VISION PRETRAINING

**Cluster-based Prediction Unit.** We propose a novel approach by grouping spatially adjacent patches into larger clusters to serve as the prediction units (see Figure 3(e)). Furthermore, our method introduces an innovative cluster-masked strategy (see Figure 3(f)), which significantly enhances the model's capability to capture local features effectively. (Please refer to Appendix B.1 for a detailed clarification of the "Cluster" terminology.)

**Cluster Formation:** See Fig 3(e); in forming clusters, we consider each patch in the image as a basic unit and group them based on their spatial proximity to ensure continuity and coherence within clusters. The size of the clusters can be adjusted according to the requirements of specific tasks or datasets, thereby reducing sequence length and computational costs. For example, in a 224x224 pixel image, using a patch size of 16x16 converts it into 196 patches; these patches can then be further combined into fewer larger clusters. This mechanism not only enhances the model's effectiveness in learning local structures but also optimizes image scanning efficiency. The clustered input $X = \{c_1, ..., c_n\}$ aims to be optimized by:

$$\mathcal{L}_{\text{MAL}} = \sum_{i=1}^{n-1} l(f([c_1, \dots, c_i]), c_{i+1}), \quad l(\hat{y}, y) = |\hat{y} - y|^2. \tag{3}$$

Here, $c_i$ represents a cluster unit formed by combining multiple adjacent image patches. For example, if the image is 12×12 patches and each cluster consists of 4×4 patches, this will generate 9 clusters. Our ablation studies (Table 6) show that using clusters as prediction targets significantly enhances performance compared to using individual pixels or patches. Next, we explore the strategies for sequencing these clusters into a coherent visual sentence.

**Cluster-Masked Generation** After serializing images into clusters, we employ a cluster-masked masking strategy (see Fig.3(f)). This approach leverages the clustered image sequences, enabling the mask to adaptively focus on both preceding and succeeding clusters. By doing so, each cluster can attend to its relevant contextual clusters. This strategy enhances the model's ability to learn from rich contextual information provided by the clustered representations.

**The importance of Cluster-Masked Masking for Local Feature Capture is seen in Appendix Section B.3.**

### 3.2.3 CLUSTER-BASED FLOW SCANNING

This section introduces a novel Cluster-Based Flow Scanning algorithm that systematically defines the sequence order for autoregressive modeling in 2D images, ensuring optimal processing of clustered representations. This algorithm enhances the efficiency of the scanning process while preserving the spatial relationships within the visual data.

Unlike the clear sequence order for autoregressive modeling in 1D sentences in NLP, 2D images require defining the sequence order when converting them into 1D visual sentences. As shown in Figure 4, we explore four primary prediction orders for arranging clusters into a sequence: 1) *Cluster-row and Horizontal Forward* (see Figure 4(a)): Process clusters by row, from the first cluster to the last cluster in each row, and scan within each cluster in a row-wise manner. 2) *Cluster-column and Horizontal Forward Scanning* (see Figure 4(b)): Scan within each cluster in a column-wise manner while processing clusters by row, from the first cluster to the last cluster in each row. 3) *Cluster-row and Horizontal Backward Scanning* (see Figure 4(c)): Similarly process clusters by row, but scan within each cluster in reverse row order, starting from the last cluster in each row. 4) *Cluster-column and Horizontal Backward Scanning* (see Figure 4(d)): Scan within each cluster in reverse column order, while processing clusters by row, starting from the last cluster in each row. Additionally, a *Random* permutation of cluster order (Figure 4(e)) was tested to avoid predefined sequential biases.

## 3.3 PARALLEL ENCODER AND DECODER ARCHITECTURE

We design a parallel encoder-decoder architecture where the encoder and decoder do not share weights(see Fig. 1). During pretraining, the encoder learns contextual information from visible positions using a Cluster-Masked approach, while the decoder reconstructs the image from the latent representation with position embeddings.

### 3.3.1 IMAGE SERIALIZATION WITH CLUSTERS

Following the ViT approach ViT Dosovitskiy et al. (2021), we first split the 2D image $x \in \mathcal{D}$ into patches, and the image patches are flattened into vectors $\{x_i\}_{i=1}^N$, where $N$ is the number of patches. Then, the vectors are linearly projected to obtain patch embeddings $\boldsymbol{W}x_i \in \mathbb{R}^D$, where $\boldsymbol{W}$ is a learnable weight matrix and $D$ is the embedding dimension. Finally, we add learnable positional embeddings $\boldsymbol{E}_{pos} = [e_1, e_2, \cdots, e_N]$ to patch embeddings, where $\boldsymbol{E}_{pos} \in \mathbb{R}^{N \times D}$. These positional embeddings are learned during the training process and provide information about the position of each patch within the original image. Thus, we obtain the initialized sequence $\boldsymbol{s} = [s_1, s_2, \cdots, s_N] = [\boldsymbol{W}x_1, \boldsymbol{W}x_2, \cdots, \boldsymbol{W}x_N] + \boldsymbol{E}_{pos}$, which serves as the input to the subsequent layers of the model.

We enhance the traditional image serialization process by implementing a clustering mechanism, which significantly improves the model's ability to capture local features and computational efficiency.

### 3.3.2 ENCODER

As illustrated in Figure 1, the encoder of our model employs an xLSTM architecture comprising $M$ layers. Each layer performs a traversal of the input sequence using a cluster-based flow scanning mechanism. This scanning mechanism enhances the model's capacity to capture dependencies across various segments of the input. Computationally, we define $h_i^{(m)}$ as the output of the $m$-th encoder layer, where $i$ is the token index. The initialized sequence $\boldsymbol{s}$ is used as the input of the first encoder layer, i.e., $h_i^{(0)} = s_i$. The forward process of the encoder can be described as follows:

$$h_{z_t}^{(m)} = \text{xLSTM}(h_{z_t}^{(m-1)}; \theta_e^{(m)}); \text{where } 1 \le m \le M \tag{4}$$

Where $\theta_e^{(m)}$ represents the parameters of the $m$-th encoder layer.

### 3.3.3 DECODER

We adopt a lightweight decoder composed of standard attention blocks, a design choice inspired by the effectiveness of asymmetric encoder-decoder architectures in masked image modeling He et al. (2022). The rationale is twofold: 1) The reconstruction task is simpler than the representation learning of the encoder, so a less complex architecture is sufficient and more computationally efficient. 2) This asymmetry forces the encoder to learn a more robust latent representation, as it cannot rely on shared architectural biases. As depicted in Figure 1, our decoder consists of $N$ layers of attention blocks followed by an MLP layer to project the reconstructed signal back to its original dimension.

During the decoding process, a lower triangular matrix is used to maintain the autoregressive nature of the model. For a sequence of length $N$, we generate a lower triangular matrix $M$ of size (N, N), where each element $\mathbf{M}_{ij}$ is defined as follows:

$$\text{content\_mask}_{ij} = \begin{cases} 0 & \text{if } i < j, \\ -\infty & \text{if } i \geq j. \end{cases} \tag{5}$$

Here, $\text{content\_mask}_{ij} = 0$ allows the $i$-th token to attend to the $j$-th token, while $\text{content\_mask}_{ij} = -\infty$ prevents it. This ensures each token attends only to itself and preceding tokens, preserving the model's autoregressive properties.

We define $g_i^{(n)}$ as the output of the $n$-th decoder layer. The position embeddings $\boldsymbol{E}_{pos}$ and the output of the last encoder layer $h_{z_t}^{(M)}$ are used as the input to the first decoder layer, i.e., $g_i^{(0)} = e_i + h_{z_t}^{(M)}$. The forward process of the decoder can be described as follows:

$$g_{z_t}^{(n)} = \begin{cases} \text{Attention}(\text{QKV} = g_{z_t}^{(n-1)}; \text{mask} = \text{content\_mask}; \theta_d^{(n)}), & \text{if } 1 \leq n < N \\ \text{MLP}(g_{z_t}^{(n-1)}; \theta_d^{(n)}), & \text{if } n = N \end{cases} \tag{6}$$

Here, $\theta_d^{(n)}$ are the parameters of the $n$-th decoder layer, which are distinct from the encoder parameters $\theta_e^{(m)}$. The number of decoder layers, $N$, determines the depth of the decoder. The output of the last decoder layer, $g_{z_t}^{(N)}$, is used to compute the loss.

### 3.4 PRETRAIN AND FINE-TUNING

Figure 2(a): Perform autoregressive pre-training on MAL using the ImageNet-1K dataset. The input image serialization uses the Cluster-Masked strategy, corresponding to Figure 3(f). The MAL consists of xLSTM encoders and decoders, followed by MLP layers that project the reconstructed signal back to its original dimension.

Figure 2(b) shows the fine-tuning stage, Cls represents classification. Where the MAL model was fine-tuned for classification tasks using the ImageNet-1K dataset. The input image sequence uses a cluster-based strategy, corresponding to Figure 3(e), without using the Masked strategy for the Cluster. The encoder uses the xLSTM blocks obtained from the pre-training stage, and the first and last patches outputted by the encoder are input to the linear classification head for classification tasks.

## 4 EXPERIMENTS

### 4.1 IMPLEMENTATION DETAILS

**Pretraining.** We pretrain MAL using the ImageNet-1K dataset Deng et al. (2009b), which contains 1.3M training images and 50K validation images where each image belongs to one of 1000 classes. Specifically, MAL-Base and MAL-small are pre-trained for 800 epochs, and MAL-Tiny is pre-trained for 400 epochs using eight NVIDIA A100 80G GPUs. We use a batch size of 2048/1024/512 for MAL-T/S/B, respectively, and a learning rate of lr = 1.5e-4$\times \frac{\text{batchsize}}{256}$. We adopt the AdamW Loshchilov & Hutter (2019) optimizer with a weight decay of 0.05. We use random resized cropping and random horizontal flipping. The pretraining input size is set to $192 \times 192$ (see Fig.2(a)).

Table 1: Performance comparison on ImageNet-1K (all image sizes are $224^2$).

| Model | Token Mixer | Param. (M) | Throughputs (imgs/s) | Top-1 (%) |
|---|---|---|---|---|
| *Tiny-size models* | | | | |
| DeiT-T | Attention | 6 | 3540 | 72.2 |
| DeiT-II-T | Attention | 6 | 3478 | 73.5 |
| DeiT-III-T | Attention | 6 | 3491 | 76.2 |
| VRWKV-T | Attention | 6 | 3640 | 75.1 |
| Vim-T | Mamba | 7 | 3178 | 76.1 |
| Mamba$^®$-T | Mamba | 9 | 3877 | 77.4 |
| ViL-T | xLSTM | 6 | 3953 | 78.3 |
| MAL-T | xLSTM | 6 | 4108 | **78.8** |
| *Small-size models* | | | | |
| DeiT-S | Attention | 22 | 2253 | 79.8 |
| DeiT-II-S | Attention | 22 | 2134 | 80.7 |
| DeiT-III-S | Attention | 22 | 2175 | 81.4 |
| VRWKV-S | Attention | 24 | 2316 | 80.1 |
| Vim-S | Mamba | 26 | 2057 | 80.5 |
| Mamba$^®$-S | Mamba | 28 | 2467 | 81.1 |
| ViL-S | xLSTM | 23 | 2515 | 81.5 |
| MAL-S | xLSTM | 23 | 2614 | **82.5** |
| *Base-size models* | | | | |
| DeiT-B | Attention | 86 | 1073 | 81.8 |
| DeiT-II-B | Attention | 86 | 1024 | 82.7 |
| DeiT-III-B | Attention | 86 | 1057 | 83.7 |
| ConvNeXt-B | Conv | 87 | 1054 | 82.0 |
| VRWKV-B | Attention | 94 | 1103 | 82.0 |
| ARM-B | Mamba | 85 | 1159 | 83.2 |
| Vim-B | Mamba | 98 | 890 | 81.9 |
| VMamba-B | Mamba | 89 | 315 | 83.9 |
| Mamba$^®$-B | Mamba | 99 | 1175 | 82.9 |
| ViL-B | xLSTM | 89 | 1198 | 82.4 |
| MAL-B | xLSTM | 89 | 1245 | **84.3** |

Table 2: Robustness and generalization evaluation on out-of-domain datasets.

| Method | IN-1K ↑ | IN-Real ↑ | IN-Adv.↑ | IN-Ren.↑ | IN-Ske.↑ |
|---|---|---|---|---|---|
| Vim-T | 76.1 | 85.4 | 9.6 | 38.8 | 26.9 |
| Vim-S | 80.5 | 86.0 | 19.7 | 45.8 | 32.5 |
| Vim-B | 81.9 | 86.2 | 27.5 | 46.0 | 33.9 |
| ViL-T | 78.3 | 85.8 | 15.2 | 42.2 | 30.0 |
| ViL-S | 81.5 | 86.5 | 23.8 | 47.6 | 35.2 |
| ViL-B | 82.4 | 87.1 | 30.9 | 48.2 | 39.0 |
| MAL-T | 78.8 | 86.4 | 16.1 | 43.4 | 31.2 |
| MAL-S | 82.5 | 87.6 | 25.3 | 48.3 | 36.3 |
| MAL-B | 84.3 | 88.4 | 32.2 | 49.2 | 40.3 |

Table 3: Results of detection and instance segmentation.

| Method | FLOPs | #Param. | AP$^b$ | AP$^m$ |
|---|---|---|---|---|
| DeiT-T | 93.5G | 8M | 41.4 | 37.9 |
| VRWKV-T | 67.5G | 8M | 41.7 | 38.0 |
| Vim-T | 64.4G | 9M | 42.7 | 38.7 |
| ViL-T | 63.2G | 8M | 42.8 | 39.0 |
| MAL-T | 60.2G | 8M | 43.5 | 39.7 |
| DeiT-S | 198.4G | 27M | 44.2 | 39.6 |
| VRWKV-S | 187.6G | 29M | 44.8 | 40.2 |
| ViM-S | 172.1G | 31M | 44.7 | 40.4 |
| ViL-S | 166.5G | 28M | 44.9 | 40.7 |
| MAL-S | 162.3G | 28M | 46.0 | 41.5 |

**Finetuning.** Following pretraining, we finetune the MAL models on the ImageNet classification task (see Fig.2(b)). Specifically, we finetune all models for 200 epochs with a batch size of 1024, with the input size set at $224 \times 224$. We use the same data augmentation as MAE He et al. (2022). We adopt AdamW as the optimizer, using a cosine decay schedule and a warm-up period of 5 epochs. Additionally, we employ the exponential moving average (EMA) Izmailov et al. (2018) for stronger performance. **More details on experiments can be found in the appendix Section Implementation Details.**

## 4.2 MAIN RESULTS

In Table 1, we compare our MAL with Attention-based DeiT, various Mamba architectures, and xLSTM-based Vision-LSTM. Our base-size MAL model achieves an accuracy of 84.3%, the highest among all models. Additionally, MAL surpasses Vim-B by 2.4% and ViL-B by 1.9%.

The table presented in the text compares different image classification models on the ImageNet-1K dataset, focusing on their performance in terms of parameter count, throughput, and top-1 accuracy across various model sizes: tiny, small, and base. The models utilize different token mixers, including attention-based architectures (like DeiT-III Touvron et al. (2022) and VRWKV Duan et al. (2024b)), Mamba architectures (such as Vim and Mamba$^®$), and xLSTM-based architectures (like ViL).

In the tiny-size category, MAL-T, an xLSTM-based model, achieves the highest top-1 accuracy of 78.8%, outperforming other models like Vim-T Wang et al. (2024) and Mamba$^®$-T Wang et al. (2024), both of which are based on Mamba architecture. Notably, MAL-T also maintains a high throughput of 4108 images per second, demonstrating its efficiency. In the small-size category, MAL-S continues to lead with an accuracy of 82.5%, surpassing attention-based models such as DeiT-III-S and VRWKV-S. Despite having a similar parameter count to its counterparts, MAL-S offers better performance and efficiency. MAL-B achieves a top-1 accuracy of 84.3% for base-size models, a significant improvement over other models like ViL-B Alkin et al. (2024b) and VRWKV-B. Additionally, MAL-B offers higher throughput, with a processing rate of 1245 images per second.

Overall, the table highlights the competitive performance of xLSTM-based models, particularly the MAL variants, across different model sizes. These models achieve high accuracy and maintain efficient throughput, making them a strong choice for image classification tasks on the ImageNet-1K dataset.

Table 4: Top-1 Accuracy Comparison of ViT, Vim, and ViL with Cluster-Masked Strategy and MAE Pretraining

| Method | Accuracy(Base) | +(Cluster-Masked) | +(MAE) |
|--------|----------------|-------------------|--------|
| ViT-B | 77.9 | 80.6 (+2.7) | 79.1 |
| Vim-B | 81.9 | 83.5 (+1.6) | 82.7 |
| ViL-B | 82.4 | 84.3 (+1.9) | 83.1 |

Table 5: Pretraining Accuracy Comparison Across Masking Strategies and Masking Ratios

| Masking Strategy | Masking Ratio | | | | |
|------------------|------|------|------|------|------|
| | 1% | 10% | 20% | 30% | 50% |
| Pixel-Masked | 81.2 | 81.2 | 81.7 | 81.5 | 81.1 |
| Patch-Masked | 81.8 | 82.3 | 82.6 | 82.5 | 82.1 |
| Cluster-Masked | 82.6 | 83.3 | **84.3** | 83.7 | 83.0 |

## 4.3 ROBUSTNESS AND GENERALIZATION

Further, we evaluate model robustness on various out-of-domain ImageNet variants (see in Table 2). Including natural adversarial examples (ImageNet-A Hendrycks et al. (2021b)), ImageNet-Ren Hendrycks et al. (2021a), image sketches (ImageNet-S Wang et al. (2019)), and ImageNet-Real Beyer et al. (2020).

In our analysis of model robustness on out-of-domain ImageNet variants, the xLSTM architectures exhibited significant performance enhancements. The MAL models, in particular, consistently outperformed their supervised counterparts, ViL, across various benchmarks such as ImageNet-Real, ImageNet-A, ImageNet-R, and ImageNet-S. For example, MAL-T surpassed ViL-T with improvements between 0.5% and 1.2% on these datasets. Additionally, MAL-S demonstrated even larger gains, with performance increases ranging from 0.7% to 1.5% compared to ViL-S. Our largest model, MAL-B, maintained this upward trend by achieving an average performance advantage of 2.33% over ViL-B, highlighting the robustness benefits of scaling up model size. These findings are comprehensively presented in Table 2, which compares different models' robustness and generalization abilities on out-of-domain datasets.

## 4.4 DETECTION AND INSTANCE SEGMENTATION

We utilize a Masked R-CNN backbone from the MMDetection Chen et al. (2019) and MMSegmentation Contributors (2020) libraries to conduct additional experiments for detection and instance segmentation. As shown in Tab. 3. $AP^b$ and $AP^m$ denote box AP and mask AP. FLOPs are calculated with an input size of $1280 \times 800$. MAL demonstrates superior performance in both box and mask precision ($AP^b$ and $AP^m$) across different training schedules. MAL-T/S achieves object detection $AP^b$ of 43.5% / 46.0%, outperforming ViL-T/S by 0.7% / 1.1% $AP^b$ and Vim-T/S by 0.8% / 1.3% $AP^b$, respectively. MAL-T/S achieves instance segmentation $AP^m$ that exceed ViL-T/S by 0.7% / 0.8% $AP^m$ and Vim-T/S by 1.0% / 1.1% $AP^m$, respectively. MAL achieves state-of-the-art performance in both object detection $AP^b$ and instance segmentation $AP^m$, with its Tiny and Small models significantly outperforming counterparts like ViL-T/S and Vim-T/S while maintaining lower computational costs (FLOPs).

## 4.5 CLUSTER-MASKED STRATEGY ON TRANSFORMER AND MAMBA

In Table 4. To validate that our proposed cluster-masked strategy enhances not only xLSTM-based models but also other visual models, we conducted experiments on two classic architectures: Vision Transformer (ViT) and Vision Mamba (Vim). All models were pretrained on the ImageNet-1K dataset and evaluated on the same validation set. We selected ViT-B and Vim-B as baseline models and implemented the cluster-masked strategy, which included image serialization into patches, cluster creation, and the application of masking methods. All models used the AdamW optimizer with a learning rate adjusted for batch size, and the input size was set to 192×192 pixels. After pretraining, the models were fine-tuned and tested on the ImageNet validation set using standard data augmentation and a cosine annealing learning rate schedule. Our results show that: In the ViT-B model, the cluster-masked strategy improved top-1 accuracy from 77.9% to 80.6%, indicating significant performance enhancement. The Vim-B model also benefited, with top-1 accuracy increasing from 81.9% to 83.5%, surpassing its previous best. In summary, this experiment confirms the effectiveness of the cluster-masked strategy across different visual model architectures.

**Extended Evaluations.** Comprehensive evaluations on semantic segmentation, High-Resolution images, and comparative studies with MAE are detailed in Appendix Section Additional Experiments, further validating MAL's generalizability across domains.

Table 6: Ablation on the number of prediction units.

| Num of Prediction unit | Cluster size | Top-1 (%) |
|---|---|---|
| 0 (Supervised) | N/A | 81.7 |
| 144 | 1×1 | 82.7 |
| 4 | 6×6 | 83.4 |
| 9 | 4×4 | **84.3** |
| 16 | 3×3 | 83.5 |
| 36 | 2×2 | 83.2 |

Table 7: Impact of prediction and scanning orders on model performance.

| Scanning Direction | Accuracy (%) |
|---|---|
| Random | 81.4 |
| Cluster-row and Horizontal Forward | 83.6 |
| Cluster-row and Horizontal Backward | 83.4 |
| Cluster-column and Horizontal Forward | 83.6 |
| Cluster-column and Horizontal Backward | 83.5 |
| Alternating & All Directions | **84.3** |

## 4.6 ABLATION STUDY

This section provides different ablations on MAL. Unless otherwise specified, all ablation studies are performed on MAL-B under 800 epochs pretraining.

**Number of Prediction Units.** Table 6 presents an ablation study on the number of prediction units used in our model. We begin with a cluster size equivalent to the patch size, resulting in a total of 144 prediction units. The results indicate that autoregressive pretraining successfully enhances the performance of the xLSTM model from 81.7% (achieved through supervised training) to 82.7%. As we progressively group multiple patches into a single cluster, thereby reducing the total number of prediction units, we observe an initial increase in performance followed by a decline. The optimal performance is achieved when the number of prediction units is set to 9, corresponding to a cluster size of 4×4. Specifically, this configuration yields a 2.6% improvement over the supervised counterpart and a 1.6% enhancement compared to the autoregressive pretraining with a 1×1 cluster size (144 prediction units).

**Analysis of Cluster Flow Scanning Orders.** Table 7 details the impact of different scanning orders on model performance. Notably, compared to a random scanning order which serves as a baseline (81.4% accuracy), even a unidirectional Cluster Flow Scan significantly improves accuracy by over 2 absolute points to 83.6%. Our proposed "Alternating & All Directions" strategy, which leverages multiple scanning patterns, further boosts performance to a peak of **84.3%**. These results strongly demonstrate that the proposed Cluster Flow Scanning Orders methodology effectively enhances model performance.

**Comparison of Masking Strategies in Pretraining.** We use the Cluster-Masked Strategy and MAE Strategy on the ViT, Vim, and ViL models respectively to demonstrate the effectiveness of our Cluster-Masked Strategy. The results in Table 4 demonstrate that the Cluster-Masked Strategy significantly outperforms the MAE Strategy. We compared three masking strategies: pixel-masked, patch-based masked, and our proposed cluster-masked method. For each strategy, we experimented with different masking ratios. The pretraining sequence length was set to 144 tokens, and we masked 1 token (1%), 14 tokens (10%), 28 tokens (20%), 43 tokens (30%), and 72 tokens (50%). We recorded the results of fine-tuning on the ImageNet-1K classification task to evaluate the effectiveness of each strategy. Table 5 summarises the results, highlighting the importance of selecting an appropriate masking ratio and strategy for effective autoregressive pretraining.

## 5 CONCLUSION

In this paper, we introduced the MAL framework, which enhances xLSTM's capabilities in visual representation learning through innovative methodologies. Our key contributions include a novel cluster-masked masking strategy that optimizes local feature capture and a new cluster-masked scanning method that improves image scanning efficiency. For the first time, we also demonstrated the application of xLSTM to autoregressive tasks in visual representation learning, showcasing its potential for handling complex visual data. Experimental results show that our methods outperform traditional supervised models while effectively leveraging the scalability of xLSTM. By addressing the limitations of previous approaches and incorporating advanced pretraining strategies, we establish a new benchmark for visual task performance. Our findings underscore the transformative potential of combining autoregressive techniques with innovative masking strategy, paving the way for future research in this area.

## ETHICS STATEMENT

This research aims to advance the field of visual representation learning. We primarily utilize publicly available and widely used academic datasets (ImageNet-1K, ADE20K, Synapse), for which ethical approvals were obtained by their original creators. We did not collect new data or involve human subjects. We acknowledge that models trained on large-scale datasets may inherit and potentially amplify societal biases present in the data. While our work does not directly address bias mitigation, we recognize it as an important direction for future research. Furthermore, like any powerful generative or representation learning model, this technology could be applied to unforeseen applications; our goal is to provide it to the research community for positive advancements.

## REPRODUCIBILITY STATEMENT

To ensure the reproducibility of our results, we provide comprehensive details throughout the paper and its supplementary materials.

- **Hyperparameters:** All hyperparameters for pre-training and fine-tuning are detailed in Section D (Table 11). We also provide an ablation study on the decoder design in Table 12.
- **Datasets:** We use publicly available datasets: ImageNet-1K Deng et al. (2009b), ADE20K Zhou et al. (2019), and the Synapse multi-organ segmentation dataset. We follow standard data processing procedures as described in the respective original works.
- **Code:** We will release our complete source code, including pre-trained model weights, upon publication. An anonymized version of the code has been submitted as part of the supplementary materials to allow for verification during the review process.
- **Infrastructure:** As detailed in Section D.2, all experiments were conducted on NVIDIA A100 GPUs.

We believe these resources are sufficient for the research community to reproduce our findings.

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

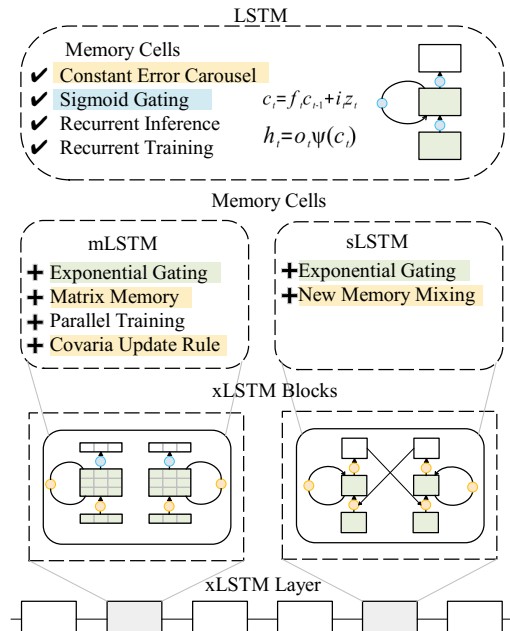

Figure 5: xLSTM architecture.

# A    xLSTM BLOCK

## A.1    sLSTM BLOCK

The sLSTM Block extends the traditional LSTM architecture by introducing exponential gating and normalization states to enhance the control over information storage and flow (see Fig.5).

**Memory Cell and State Update**    The memory cell is updated as follows:

$$C_t = f_t C_{t-1} + i_t z_t,$$
$$Z_t = tanh(W_z x_t + R_z h_{t-1})',$$

(7)

where $c_t$ is the memory cell at time step t, $f_t$ and $i_t$ are the forget and input gates, respectively, $z_t$ is the candidate memory, controlled by $W_z$, and $R_z$. Normalization state update:

$$n_t = f_t n_{t-1} + i_t,$$

(8)

Where $n_t$ is the normalization state that balances the contribution of the forget and input gates. The hidden state is computed as:

$$h_t = \frac{o_t c_t}{n_t},$$
$$o_t = \sigma(W_o x_t + R_o h_{t-1})',$$

(9)

Where $o_t$ is the output gate, controlling the final output $h_t$. Normalizing $c_t$ by $n_t$ ensures numerical stability.

**Projection and Residual Connection**    The hidden state is further processed through up projection, non-linear transformation, and down-projection:

$$
\begin{aligned}
y_{left} &= W_{up-left}h_t, \\
y_{right} &= W_{up-right}h_t, \\
y_{gated} &= GELU(y_{right}), \\
y_{out} &= W_{down}(y_{left} \cdot y_{gated}),
\end{aligned}
\tag{10}
$$

The final output includes a residual connection:

$$
F = y_{out} + x,
\tag{11}
$$

Where F represents the final output.

## A.2 mLSTM Block

The mLSTM Block enhances memory capacity by transforming the memory cell from a scalar to a matrix, which allows for more intricate storage and representation.

**Matrix Memory and Key-Value Storage**    The memory cell is updated in matrix form:

$$
\begin{aligned}
C_t &= f_t C_{t-1} + i_t v_t k_t^T, \\
k_t &= \frac{1}{\sqrt{d}} W_k x_t + b_k, \\
v_t &= W_v x_t + b_v,
\end{aligned}
\tag{12}
$$

where $C_t$ is represents the memory matrix, while $v_t$ and $k_t$ denote the value and key vectors, respectively. The weights for generating the key and value vectors are represented by $W_k$ and $W_v$, while the corresponding biases are denoted as $b_k$ and $b_v$.

**Memory Retrieval and Normalization**    To extract information from the memory matrix, the mLSTM Block utilizes a query vector $q_t$:

$$
\begin{aligned}
h_t &= o_t \odot \frac{C_t q_t}{max(|n_t^T q_t, 1|)}, \\
q_t &= W_q x_t + b_q, \\
n_t &= f_t n_{t-1} + i_t k_t,
\end{aligned}
\tag{13}
$$

where $q_t$ is the query vector and $n_t$ is the normalization state.
The matrix-based memory update and the retrieval mechanism significantly enhance the model's ability to capture complex temporal relationships. By effectively managing and retaining information across time steps, it preserves long-term dependencies. This dynamic memory adjustment improves the model's capacity to model intricate temporal patterns, which is crucial for tasks involving sequential data, where understanding complex dependencies is essential for accurate modeling.

# B THEORETICAL ANALYSIS AND CLARIFICATIONS

## B.1 TERMINOLOGY CLARIFICATION: "CLUSTER" VS. "GROUPED PATCHES"

We acknowledge that the term "Cluster" might typically imply the result of an unsupervised clustering algorithm (e.g., K-Means). To avoid ambiguity, we clarify our terminology as follows:

- **Definition:** In the context of MAL, a "Cluster" refers to a deterministic grouping of spatially adjacent patches (e.g., a $2 \times 2$ grid of patches forms one cluster). This can be conceptually understood as "Macro-patches" or "Two-level Patch Modeling".

- **Motivation:** We use this terminology to emphasize the semantic grouping capability. Unlike individual patches which often contain only texture or edges, a "Cluster" covers a larger receptive field, encapsulating semantically meaningful parts of an object (e.g., an entire dog's ear vs. just fur texture).

## B.2 INTUITIVE ANALYSIS: WHY MAL OUTPERFORMS STANDARD MIM?

While standard Masked Image Modeling (MIM) and MAL both involve masking and reconstruction, MAL achieves superior performance due to two key factors designed for the xLSTM architecture:

**1. Task Difficulty: From Texture Completion to Structure Reconstruction.** Predicting a single $16 \times 16$ patch (as in standard approaches) is often a low-level texture completion task. The model can rely on immediate pixel continuity. In contrast, MAL predicts an entire "Cluster" (e.g., $4 \times 4$ patches) at once. This significantly increases the task difficulty, forcing the model to understand high-level geometric structures and semantic parts rather than just local textures.

**2. Sequence Abstraction for xLSTM Memory.** xLSTM is a recurrent model with memory states. A standard patch sequence (e.g., 144 steps) is long and contains redundant information, which can saturate the memory with low-level details. By grouping patches into clusters, MAL reduces the sequence length (e.g., to 9 steps) and increases the "semantic jump" between steps. This abstract sequence structure aligns perfectly with xLSTM's memory mechanism, allowing it to capture long-range dependencies more effectively without being overwhelmed by local redundancies.

## B.3 IMPORTANCE OF CLUSTER-MASKED MASKING FOR LOCAL FEATURE CAPTURE

The importance of cluster-masked masking lies in its facilitation of information exchange within local regions, enabling the model to better understand the relationships and boundaries within objects. Compared to traditional fully connected masking or random masking, cluster-masked masking allows the model to focus more on local details, which is particularly crucial for visual tasks that require precise boundaries, such as semantic segmentation. Moreover, since clusters contain multiple adjacent patches, this design helps smooth out noise and enhances robustness, especially when dealing with complex backgrounds or low-resolution images.

# C ADDITIONAL EXPERIMENTS

## C.1 SEMANTIC SEGMENTATION

**Settings.** We conduct experiments for semantic segmentation on the ADE20K Zhou et al. (2019) and use UperNet Xiao et al. (2018) as the segmentation framework.

**Results.** As shown in Tab. 8, MAL consistently outperforms ViL across different scales: 0.7 mIoU higher for MAL-T over ViL-T, and 1.2 mIoU higher for MAL-S over ViL-S. Compared to the ResNet-101 backbone, our MAL-S achieves better segmentation performance with nearly $2\times$ fewer parameters.

## C.2 COMPARISON WITH MAE METHOD

Moreover, we extended our evaluation by applying the MAE method to detection and instance segmentation tasks. According to the table 9, the Cluster-Masked method outperforms MAE across all models in both object detection $AP^b$ and instance segmentation $AP^m$ tasks, achieving performance improvements of over 1% in every case. This demonstrates that the clustering-based masking strategy can more effectively capture semantic information in images, thereby enhancing generalization in downstream tasks.

## C.3 HIGH-RESOLUTION IMAGE EXPERIMENT

We validated the generalization capability of MAL in the medical imaging domain by conducting transfer learning experiments on the Synapse multi-organ segmentation dataset.

| Method | Backbone | image size | #param. | $val$ mIoU |
|---|---|---|---|---|
| UperNet | ResNet-101 | $512^2$ | 86M | 44.9 |
| UperNet | DeiT-Ti | $512^2$ | 11M | 39.2 |
| UperNet | DeiT-S | $512^2$ | 43M | 44.0 |
| UperNet | Vim-Ti | $512^2$ | 13M | 41.0 |
| UperNet | Vim-S | $512^2$ | 46M | 44.9 |
| UperNet | ViL-T | $512^2$ | 11M | 41.2 |
| UperNet | ViL-S | $512^2$ | 42M | 46.3 |
| UperNet | MAL-T | $512^2$ | 11M | 41.9 |
| UperNet | MAL-S | $512^2$ | 42M | **47.5** |

Table 8: Results of semantic segmentation on the ADE20K $val$ set.

| Method | Model | $AP^b$ | $AP^m$ |
|---|---|---|---|
| MAE | VIM-S | 43.1 | 39.2 |
| | VIL-S | 43.8 | 39.5 |
| | MAL-S | 44.5 | 40.1 |
| Cluster-Masked | VIM-S | 44.7 (+1.6) | 40.4 (+1.2) |
| | VIL-S | 44.9 (+1.1) | 40.7 (+1.2) |
| | MAL-S | 46.0 (+1.5) | 41.5 (+1.4) |

Table 9: Performance Comparison Between MAE and Cluster-Masked Methods Across Model Scales

Annotation Targets: 9 abdominal organs.

Data Specifications: Uniform image size of 512×512.

Pretraining: Used MAL-Base weights pre-trained on ImageNet-1K.

Fine-tuning Architecture: Integrated a U-Net style decoder (with skip connections and upsampling modules) after the MAL encoder.

As shown in Tab. 10, experimental results demonstrate that MAL achieved a mean DSC of 84.64% on Synapse, outperforming models like U-Net Ronneberger et al. (2015), SwinUNet Cao et al. (2021), Swin-UMamba Liu et al. (2024a) and RWKV-UNet Jiang et al. (2025), highlighting its potential for medical segmentation tasks.

## D  IMPLEMENTATION DETAILS

This section provides a comprehensive overview of our experimental setup, including the training strategy, computational costs, and specific hyperparameters used.

### D.1  PRE-TRAINING AND FINE-TUNING STRATEGY

Our methodology is a two-stage process designed to learn robust visual representations and then adapt them to specific tasks.

| Methods | DSC(%)↑ | HD95 (mm)↓ | Aorta | Gallbladder | Kidney(L) | Kidney(R) | Liver | Pancreas | Spleen | Stomach |
|---|---|---|---|---|---|---|---|---|---|---|
| UNet | 76.85 | 39.70 | 85.66 | 53.24 | 81.13 | 71.60 | 92.69 | 56.81 | 87.46 | 69.93 |
| Swin-UNet | 79.13 | 21.55 | 85.47 | 66.53 | 83.28 | 79.61 | 94.29 | 56.58 | 90.66 | 76.60 |
| VM-UNet | 82.38 | 16.22 | 87.00 | 69.37 | 85.52 | 82.25 | 94.10 | 65.77 | 91.54 | 83.51 |
| HC-Mamba | 79.58 | 26.34 | 89.93 | 67.65 | 84.57 | 78.27 | 95.38 | 52.08 | 89.49 | 79.84 |
| Swin-UMamba | 82.26 | 19.51 | 86.32 | 70.77 | 83.66 | 81.60 | 95.23 | 69.36 | 89.95 | 81.14 |
| RWKV-UNet | 84.02 | 15.70 | 89.53 | 68.94 | 87.63 | 84.07 | 95.57 | 69.38 | 90.95 | 86.09 |
| MixFormer | 82.64 | 12.67 | 87.36 | 71.53 | 86.22 | 83.19 | 95.23 | 66.82 | 89.98 | 80.77 |
| MAL | **84.64** | **10.37** | **90.47** | **74.98** | **90.57** | **85.18** | **95.84** | **71.75** | **92.85** | **86.77** |

Table 10: MAL Performance on High-Resolution Medical Image Segmentation. Here, Bold Black Data indicates the Best Result, and Underlined Black Data Denotes the Second-Best Result.

| Parameter | Value |
|---|---|
| Pretrain Epochs | 400 (Tiny), 800 (Small/Base) |
| Batch size | 2048 (Tiny), 1024 (Small) |
| | 512 (Base) |
| Model | |
|    Patch size | 16x16 |
|    Latent dimension | 192 (Tiny), 384 (Small) |
| | 768 (Base) |
|    Depth | 12 |
| Optimizer | AdamW |
|    Base Learning rate | 5e-4 |
|    Weight decay | 0.05 |
|    Momentum | $\beta_1 = 0.9, \beta_2 = 0.999$ |
| Precision | `bfloat16` |
| Learning rate schedule | cosine decay |
|    Warmup epochs | 5 |
| Train Data Augmentation | |
|    RandomResizedCrop | 192 |
|    RandomHorizontalFlip | $p = 0.5$ |
|    Normalize | ImageNet-1K statistics |

Table 11: Hyperparameters for training MAL on ImageNet-1K. The pretraining is conducted at a resolution of 192, followed by fine-tuning at 224 resolution.

**Stage 1: Image Autoregression Pre-training**  In the first stage (see Fig. 2(a)), we pre-train the MAL framework using an autoregressive objective on the ImageNet-1K dataset Deng et al. (2009a). By predicting the next image patch in a sequence, the model learns complex visual patterns and captures intricate spatial relationships.

**Stage 2: Fine-Tuning**  In the fine-tuning stage, the content mask and the decoder are removed. The pre-trained encoder is then fine-tuned for classification tasks using a linear classification head attached to the output of the first and last patches (see Fig. 2(b)).

### D.2 COMPUTATIONAL COST AND INFRASTRUCTURE

**Training Time**  All experiments were conducted on a server with eight NVIDIA A100 80G GPUs. For our main model, MAL-Base, the complete autoregressive pre-training phase for 800 epochs took approximately 56 hours. The subsequent fine-tuning stage for 200 epochs was considerably faster, finishing in approximately 14 hours.

**Memory Usage**  During the pre-training of MAL-Base with a total batch size of 512 (i.e., 64 per GPU), the peak GPU memory consumption was measured at approximately 35 GB per GPU. For inference efficiency, as shown in Table 1, MAL-B maintains a high throughput competitive with other state-of-the-art models.

### D.3 HYPERPARAMETERS AND ABLATION STUDIES

The specific hyperparameters for our training process are outlined in Table 11.

**Decoder Design.**  Our exploration into decoder design is summarized in Table 12. We first focused on the design of *decoder depth*, finding that performance saturated at a depth of 8. With an 8-layer decoder, we then studied the optimal width by comparing three options: 384, 512, 1024. We empirically observed that a decoder width of 512 yielded the best accuracy.

**Ablation Study on Decoder Architecture: xLSTM vs. Transformer vs. Mamba.**  To rigorously validate our design choice of using a lightweight Transformer-based decoder, we conducted an extended ablation study. We compared our default MAL model (xLSTM Encoder + Transformer

| Dec. Depth | Dec. Width | Top-1 (%) |
|:---:|:---:|:---:|
| 4 | 512 | 82.9 |
| 6 | 512 | 83.5 |
| 8 | 512 | **84.3** |
| 10 | 512 | 84.3 |
| 8 | 384 | 83.6 |
| 8 | 512 | **84.3** |
| 8 | 1024 | 84.0 |

Table 12: Ablation on decoder designs.

Decoder) against two variants where the decoder utilizes recurrence-based blocks: 1) **xLSTM Decoder**: Creating a symmetric architecture identical to the encoder. 2) **Mamba Decoder**: Using the Mamba SSM block to test another linear-complexity alternative.

The comparison on the ImageNet-1K dataset is presented in Table 13.

| Decoder Architecture | Token Mixer | Top-1 Accuracy (%) |
|:---|:---:|:---:|
| **Transformer (Default)** | **Attention** | **84.3** |
| Symmetric Variant | xLSTM | 84.1 |
| SSM Variant | Mamba | 84.0 |

Table 13: Ablation study on decoder architectures. Comparing the default lightweight Transformer decoder against recurrence-based decoders (xLSTM and Mamba).

**Analysis.** As shown in Table 13, employing recurrence-based blocks in the decoder resulted in a performance degradation compared to the default Transformer decoder (xLSTM: -0.2%, Mamba: -0.3%). This empirical evidence strongly supports our architectural design principles detailed in Section 3.3.3:

- **Benefit of Asymmetric Architecture:** The performance drop with the xLSTM/Mamba decoders suggests that a symmetric or overly powerful decoder may relax the learning signal for the encoder. By using a structurally distinct and lightweight Transformer decoder, we enforce a stronger asymmetry. This discrepancy prevents the model from relying on shared architectural biases (such as recurrent states) and forces the encoder to learn more robust and generalized latent representations to enable reconstruction.

- **Efficiency Mismatch:** Both xLSTM and Mamba are designed for modeling complex long-range dependencies. Applying these heavy-duty sequence modelers to the relatively simpler reconstruction task introduces architectural redundancy without performance gains.

**Conclusion.** These results confirm that the "heavy Encoder (xLSTM) + light Decoder (Transformer)" design is not only computationally efficient but also essential for maximizing the representation learning capability of the MAL framework.

## E  LIMITATIONS

One limitation of this work is the computational resource constraints, which prevented us from pretraining larger-scale xLSTM models (e.g., MAL-Huge). While our results show a consistent and positive scaling trend from the Tiny to the Base model, further research is required to fully validate the scaling properties of our approach on even larger models. We believe this is a promising direction for future exploration.

## F  LLM USAGE STATEMENT

During the preparation of this work, the authors used a large language model (LLM) solely for the purpose of grammar checking and improving the readability of the text. The LLM was not used for generating core ideas, experimental results, or conclusions.

