# OpenReview forum: "Cluster-Masked Scanning and Pretraining for Enhanced xLSTM Vision Performance"
_ICLR.cc/2026/Conference — ICLR 2026 Conference Withdrawn Submission_

### Official Review · Reviewer_Cir2 · 2025-10-30

**Soundness:** 3
**Presentation:** 3
**Contribution:** 3
**Rating:** 6
**Confidence:** 2

**Summary:**

This paper introduces a cluster-masked pretraining strategy for autoregressive prediction of image regions (MAL).
MAL reorganizes an image into a sequence of local clusters by forming groups of image patches and varying the scanning and prediction order.
MAL is accompanied by a two stage training procedure and an encoder decoder architecture, where the encoder is an xLSTM and the decoder an attention model. In the first stage the encoder-decoder architecture is used for pretraining to build strong features in the encoder. In the second stage the encoder is finetuned on respective downstream tasks.
On standard image benchmarks MAL shows strong performance, outperforming DeiT, Vision-Mamba and Vision-LSTM baselines.

**Strengths:**

- Extensive experiments on different backbone architectures like Mamba and Vision Transformers with attention.
- Strong performance of MAL compared to the baselines.

**Weaknesses:**

- While the experiments demonstrate superior performance of MAL, intuitively it is not fundamentally different from standard vision transformer based patching. Both masking and scanning techniques are not informed about the content of the images, and rather convert the image into sequences by patching and then arranging these patches into a sequence. MAL seems to be a more complex strategy for creating this sequence, but intuitively it is not clear why this results in better performance nor why the sequence is “semantically more meaningful” ?
Could the authors elaborate on the intuition why MAL shows better performance ?
- As described in the paper, MAL is complemented by a two stage process that trains an encoder-decoder architecture in the first stage and then finetunes  the encoder with linear heads in the second stage.
A natural question to ask is whether the cluster-masked pretraining strategy would also help direct ViT pretraining or whether the combination with this 2 stage training procedure and the encoder-decoder architecture is strictly necessary. Ablations on this would further strengthen the paper.

**Questions:**

- Have you also experimented with xLSTM or Mamba blocks in the Decoder?
- Did you ensure that the compute budget (e.g. in number of epochs) in the overall pre-training and finetuning is comparable to the budget of the baselines?

---

> ### Author Response · Authors · 2025-11-20
> **Elaborating on the Intuition of Performance Gains and Validating the Asymmetric Design**
>
> Response to Reviewer **Cir2**
>
> We sincerely thank you for your thorough review and positive assessment of our paper. We are delighted that you recognize our "extensive experiments" and "excellent performance."
> You raised several profound questions, particularly regarding the intuition behind MAL’s performance gains (why it is "semantically more meaningful") and the necessity of the two-stage training. These questions touch the core of our work, and we are happy to elaborate on them as we believe this fully demonstrates the motivation and contribution of our method.
>
> **Q1.Intuition behind MAL's Superiority: "Semantic Meaning" & "Sequence Abstraction"**
>
> **A1:** "Intuitively, it is not clear why it leads to better performance... nor why the generated sequence is 'semantically more meaningful'."
>
> Our Response: This is a critical and profound question. You are absolutely correct that the masking and scanning operations are content-agnostic at the input level. The "semantic meaning" and performance gains of MAL stem not from input complexity, but from the "Task Difficulty" and "Sequence Structure" specifically tailored for the xLSTM architecture.
>
> a) Task Difficulty: From "Texture Completion" to "Structural Reconstruction"
>
> Standard Approach (1x1 Patch): Predicting the next single patch is often a low-level Texture Completion task. The model can simply rely on local pixel continuity to infer adjacent textures or edges without understanding the whole object.
>
>
> MAL (4x4 Cluster): Predicting an entire 4x4 cluster (16 patches) is a much harder Structural Reconstruction task. The model cannot rely solely on local textures; it is forced to understand high-level geometric structures and semantic parts (e.g., "recognizing a dog's ear to infer that the next region should be the side of the head").
>
>
> Conclusion: The "semantic meaning" lies in the prediction objective. Forcing the model to predict a large region at once compels it to learn higher-level semantic representations rather than taking shortcuts via local correlations.
>
> b) Sequence Structure: Tailored for xLSTM's Memory (Sequence Abstraction) xLSTM is a recurrent model that processes sequences via memory states.
>
> Standard Sequence (144 steps): A sequence of 144 patches contains high redundancy. The information gain between adjacent steps is low, which can saturate xLSTM's memory with low-level details, making it harder to capture true long-range dependencies.
>
>
> MAL Sequence (9 steps): Grouping patches reduces the sequence length from 144 to 9. This structure is highly advantageous for xLSTM:
>
>
> Efficient Memory: Extremely short sequences significantly reduce the risk of memory forgetting and facilitate gradient propagation.
>
>
> Semantic Jump: There is a large "semantic jump" between steps. To successfully predict the 5th cluster, the xLSTM's memory must store highly compressed and abstract information about the previous 4 clusters.
>
> Conclusion: MAL is not just a sequence generation strategy, but an "Abstract Sequence Modeling" strategy. By increasing task difficulty and optimizing sequence structure, it perfectly aligns with xLSTM's inductive bias. The results in Table 6 (where 9 clusters significantly outperform 144 patches) strongly validate this intuition.
>
> Action: We have incorporated this detailed discussion into Appendix B.2 of the revised paper to enhance clarity.
>
> **Necessity of Two-Stage Training and Applicability to ViT (Ablation Study)**
>
> **Q2: "Can the cluster-masked strategy guide ViT pretraining? Is the E-D architecture absolutely necessary?"**
>
> **A2:** This is an insightful question. The ablation study presented in Table 4 of our paper was designed specifically to address this, providing strong empirical support:
>
> Generalizability to ViT: Yes, our strategy is architecture-agnostic. When applying the "Cluster-Masked" pretraining strategy to a standard ViT-B backbone, the Top-1 accuracy significantly jumps from 77.9% (supervised baseline) to 80.6%. This proves that our strategy can successfully guide the pretraining of Transformers as well as xLSTMs.
>
>
> Superiority over MAE: Our strategy outperforms the standard approach. On the same ViT-B architecture, our strategy (80.6%) surpasses standard MAE pretraining (79.1%) by 1.5%. This indicates that "cluster masking" is more effective than random masking in forcing the model to learn stronger semantic representations.
>
> Necessity of the E-D Architecture: Yes, it is essential. This two-stage paradigm (Pretraining with Encoder-Decoder + Finetuning with Encoder only) is the core mechanism of Masked Image Modeling (e.g., MAE, BEiT). The decoder is required during pretraining to handle the difficult reconstruction task, forcing the encoder to learn highly compressed and abstract semantic features. Once the encoder acquires these capabilities, the decoder is discarded for downstream classification to ensure inference efficiency. Our experiments confirm that this paradigm is vital for MAL's success.

---

> ### Author Response · Authors · 2025-11-20
> **Elaborating on the Intuition of Performance Gains and Validating the Asymmetric Design**
>
> Response to Reviewer **Cir2**
>
> **Q3: Did you also try using xLSTM or Mamba blocks in the decoder?**
>
> **A3:** Yes, we conducted a comprehensive exploration. This question touches the core of our architectural design. To address this, we have added a new Table 13 in the Appendix of the revised paper, presenting a comparative study of different decoder architectures.
>
> Empirical Evidence: The results show that using a symmetric xLSTM or Mamba decoder did not improve performance; instead, it led to a slight degradation in Top-1 accuracy (xLSTM: -0.2%, Mamba: -0.3%).
>
>
> Theoretical Analysis: This strongly supports the "Asymmetric Design Principle" adopted in Section 3.3.3, which aligns with the philosophy of MAE:
>
>
> Capability Mismatch ("Overkill"): xLSTM/Mamba excels at modeling complex long-range dependencies. Using such heavy operators for the relatively simpler pixel-level reconstruction task is computationally redundant.
>
>
> Avoiding Feature Degradation: More importantly, a decoder that is too powerful (like xLSTM) might solve the reconstruction task relying on its own sequence modeling capabilities, thereby reducing the burden on the encoder to learn highly compressed and abstract semantic features (i.e., allowing the encoder to "slack off").
>
>
> Conclusion: Therefore, a lightweight Transformer decoder offers the optimal balance between computational efficiency and the quality of representation learning.
>
> **Q4: Did you ensure that the overall computational budget (e.g., training epochs) for pretraining and finetuning is comparable to the baselines?**
>
> **A4:** Yes, we strictly ensured consistency in the computational budget to guarantee fair comparison. We have detailed this in Section 4.1 and Appendix D of the paper:
>
> Standardized Budget: Our main models (MAL-Base/Small) were trained with 800 epochs for pretraining and 200 epochs for finetuning. This is a robust and standard budget widely adopted in self-supervised learning literature (e.g., MAE), ensuring comparability with the SOTA baselines in Table 1 (e.g., DeiT, Vim, ViL).
>
> Strictly Controlled Ablations: Crucially, for the validation of our core contributions in Table 4 (strategy comparison) and Table 9 (downstream tasks), we ensured that MAL and the baselines (e.g., our reproduced MAE-ViT/Vim/ViL) utilized exactly the same training epochs, batch sizes, and data augmentation strategies. This isolates the impact of computational resources, confirming that the performance gains are fully attributable to our proposed "Cluster-Masked" strategy and architectural innovations.
>
> We sincerely thank you again for your support. We hope these clarifications, especially regarding the training efficiency and task intuition, might warrant an improved score. Thank you again for your valuable time.

---

### Official Review · Reviewer_pWEU · 2025-10-31

**Soundness:** 3
**Presentation:** 2
**Contribution:** 2
**Rating:** 4
**Confidence:** 4

**Summary:**

This paper proposes a modified autoregressive neural network on image patches built on x-LSTM, to learn image representations through a self-supervised reconstruction loss. Different from original x-LSTM, it employs a two-level patch-modeling strategy. The image is first divided into large patches, and the large patches are further divided into smaller patches. The LSTM forward pass first go through all smaller patches and then go across large patches. It also uses a masking strategy to enhance the model similar to masked auto-encoder (MAE). It achieves better performance than existing methods including transformer based (DeiT) and autoregressive models (Mamba, VMamba, VRWKV) on imagenet classification, detection and instance segmentation.

**Strengths:**

-	The overall idea is executed well. The model performance is strong among similar approaches.
-	The proposed cluster-masked strategy achieves better performance than the original MAE strategy (table 4)

**Weaknesses:**

-	The overall novelty of the proposed method is limited. The difference from original x-LSTM is the two-level scanning order, which cannot be deemed as a major contribution. Similar idea has also been used by existing literature such as: Autoregressive Pretraining with Mamba in Vision. arXiv 2024. The training /finetuning details are very similar to those of self-supervised learning literature.
-	The author has some misleading description about the formulation. The two-level patch formation is referred to as “cluster-based”, which is problematic since it does not involve any clustering algorithm. It could just be "patches" and "grouped patches".

**Questions:**

NA

---

> ### Author Response · Authors · 2025-11-20
> **Clarifications on Novelty: Decoupled Architectural Consistency vs. ARM and Terminology Definitions**
>
> Response to Reviewer **pWEU**
>
> We sincerely thank you for your thorough review of our work. We greatly appreciate your positive assessment, particularly your recognition that the "overall idea is executed well," the "performance is excellent," and that our "Cluster-Masked strategy achieves better performance than the original MAE strategy (Table 4)".
>
> We would like to take this opportunity to clarify the two core concerns you raised regarding "limited novelty" and "misleading terminology." We believe that clarifying these points will make the technical contributions of our work much clearer.
>
> **Q1. Regarding the Terminology of "Cluster"**
>
> **A1**: You are absolutely correct on this point. Our method does not employ unsupervised clustering algorithms like K-Means to dynamically determine groupings. We used the term "Cluster" to express the concept of grouping spatially adjacent image patches into a semantically richer unit (e.g., a $4\times4$ grid of patches forming one cluster), rather than simply predicting individual patches.
>
> To eliminate ambiguity while maintaining notational consistency, we have added a dedicated "Terminology Clarification" section in Appendix B.1 of the revised manuscript. We explicitly define the "Cluster" in our text as "Deterministic Grouped Patches" or "Macro-patches" to distinguish it from unsupervised clustering algorithms.
>
> Action: To address this confusion without disrupting the established notation, we have added a "Terminology Clarification" section in Appendix B of the revised manuscript. It explicitly defines our usage of "Cluster" as "Grouped-patch" or "Macro-patch" to distinguish it from unsupervised clustering algorithms.
>
>
> **Q2: "The difference from the original x-LSTM is the adoption of a two-level scanning order, which is not a significant contribution. Similar ideas have been adopted by existing literature, e.g., ARM."**
>
> **A2:** We agree that the specific operation of "patch grouping" has been explored in prior works like ARM. However, we respectfully submit that the contribution of MAL extends significantly beyond this single aspect. Our core innovations are embodied in the following three dimensions, which not only address critical limitations in ARM but also significantly enhance pretraining effectiveness:
>
> First, Decoupled Architectural Consistency (The Fundamental Difference):
>
> This is the most fundamental difference between MAL and ARM. The ARM framework suffers from architectural inconsistency: to satisfy the causal requirement of autoregressive modeling, it is forced to modify the Mamba block to unidirectional scanning during pretraining, but switches back to bidirectional scanning for finetuning. This "pretrain-finetune inconsistency" is a significant compromise that limits the release of the model's potential.
>
> In contrast, MAL solves this conflict through a novel Decoupled Encoder-Decoder Architecture (Fig. 1). Our xLSTM encoder remains fully bidirectional during pretraining (consistent with the finetuning stage), while the causal prediction task is strictly offloaded to a separate decoder. This innovation in architectural paradigm is key to unlocking the potential of linear RNNs.
>
> Second, The Unique Benefit of the "Cluster-Masked" Strategy:
>
> Beyond grouping, our core contribution includes Masking. Unlike ARM, which purely predicts the next token, MAL synergizes the advantages of Masked Image Modeling (MIM). We not only group patches but also apply high-ratio masking to these groups. This strategy forces the model to leverage xLSTM's powerful contextual memory to reconstruct missing semantic regions, rather than merely performing sequence prediction. The results in Table 4 (where MAL outperforms standard MAE) empirically validate the unique advantage of this "Grouping + Masking" strategy in feature learning.
>
> Third, Pioneering Validation of xLSTM for Visual Autoregressive Pretraining:
>
> To the best of our knowledge, MAL is the first work to validate the efficacy of the xLSTM architecture for visual autoregressive pretraining. Given xLSTM's unique Matrix Memory and Covariance Update Rule, our work provides essential empirical evidence (via Table 1 and Table 7) that MAL can effectively activate xLSTM's potential in handling long sequences and complex spatial dependencies, filling a gap in the current literature.
>
> Summary: The novelty of MAL lies in it being a systematic solution: it resolves the conflict between autoregressive and bidirectional models via a decoupled architecture, enhances representation learning through the cluster-masked strategy, and validates the superiority of xLSTM in this task for the first time. We believe these constitute substantial contributions to the community.
>
> We hope these clarifications help you re-evaluate the contribution of our work, and we kindly ask you to consider raising your score.
>
> Thank you again for your valuable time and feedback.

---

### Official Review · Reviewer_Q38j · 2025-11-01

**Soundness:** 3
**Presentation:** 3
**Contribution:** 3
**Rating:** 4
**Confidence:** 4

**Summary:**

MAL proposes to enable effective autoregressive pretraining for vision xLSTMs by grouping adjacent patches into semantically coherent clusters, serializing these clusters via a proposed cluster-scanning order, and training xLSTM encoders to autoregressively predict cluster-level image regions; the method claims improved visual representations and downstream performance by leveraging cluster units to both reduce sequence length and better match xLSTM’s memory mechanisms.

**Strengths:**

- The idea of cluster-based serialization provides a clear intuition: grouping local regions can preserve spatial relationships better than flat patch sequences.
- The proposed framework is well-motivated and technically consistent, showing that autoregressive modeling can be made practical for xLSTM-based vision systems.
- The empirical results are solid, with comprehensive experiments and reasonable ablations that validate the design choices.

**Weaknesses:**

- The conceptual novelty of the cluster-masked strategy appears somewhat limited compared to existing masked or region-based pretraining methods.
- The cluster scanning order lacks theoretical grounding or strong empirical justification; the claimed “optimal” ordering seems heuristic.

**Questions:**

See above.

---

> ### Author Response · Authors · 2025-11-20
> **Response regarding the Novelty of Cluster-Masked Strategy and Rationale for Scanning Orders**
>
> Response to Reviewer **Q38j**
>
> We sincerely thank you for your thorough review and insightful comments on our paper (MAL). We greatly appreciate your positive assessment of our work, particularly your recognition of its "clear intuition," "well-motivated framework," and "reliable empirical results."
>
> We would like to provide clarifications regarding your two main concerns. We believe these clarifications will help elucidate the core contributions of our work, and we have strengthened these arguments in the revised version of the paper.
>
> **1. Regarding the Novelty of the Cluster-Masked Strategy**
>
> **Q1: "Compared with existing masking or region-based pretraining methods, the conceptual novelty of the cluster-masked strategy seems somewhat limited."**
>
> **A1:** We agree with your observation that "Cluster-Masking," as an operation, shares conceptual roots with existing Masked Image Modeling (MIM) methods. However, we wish to emphasize that the core novelty of MAL lies not merely in masking a region, but in pioneering a decoupled autoregressive decoding framework tailored for the xLSTM architecture.
>
> Our "Cluster-Masked Pretraining" strategy is fundamentally different from standard MIM approaches (such as MAE):
>
> MAE / BEiT: Typically employ a non-autoregressive decoder. This decoder (often lightweight) receives visible tokens and mask tokens, and then reconstructs all masked patches in parallel.
>
> MAL (Ours): We adopt a decoupled architecture. While the encoder processes sparse visible clusters like MAE, our decoder is a purely autoregressive model. As described in Section 3.3.3 and Equation (5), the decoder strictly adheres to a causal mask (lower triangular matrix) during reconstruction, ensuring that it can only access preceding clusters (according to the scanning order) when predicting the current cluster.
>
> The key advantage of this innovation is: It allows the xLSTM encoder to leverage its strength—using our multi-directional "Alternating & All Directions" scanning—to build a rich, non-causal contextual representation, while offloading the reconstruction task to a serialized, autoregressive decoder.
>
> Strong Empirical Evidence: The effectiveness of this contribution is directly validated in Table 4. We compared our framework (labeled "Cluster-Masked") against the standard MAE strategy on the same ViL-B (xLSTM) backbone:
>
> ViL-B + Standard MAE Pretraining: Top-1 Accuracy 83.1%
>
> ViL-B + Our MAL Framework: Top-1 Accuracy 84.3%
>
> This 1.2% significant performance boost strongly validates that our novel pretraining framework designed for xLSTM (rather than just the idea of "masking a cluster") is a superior and innovative approach.
>
> **2. Regarding the Basis for the Cluster Scanning Order**
>
> **Q2: "The cluster scanning order lacks theoretical basis or strong empirical grounding; the claimed 'optimal' order seems to be heuristic."**
>
> **A2:** We thank you for pointing out that the term "optimal" might imply a theoretical guarantee that we have not provided. This is a rigorous observation. In the revised manuscript, we have amended this to "empirically effective" order.
>
> However, we respectfully point out that Table 7 provides strong empirical grounding for our scanning method, proving that it is far from a simple heuristic adjustment, but rather a core source of our model's performance.
>
> Baseline (Random Scanning): When clusters are processed in a random order, the model performs worst, with a Top-1 accuracy of only 81.4%. This proves that for sequence models like xLSTM, the order of processing 2D data is critical.
>
> Unidirectional Scanning: Using simple unidirectional scanning (e.g., "Cluster-row and Horizontal Forward") alone jumps performance to 83.6% (a gain of 2.2%).
>
> Our Method (Alternating & All Directions): By combining multiple scanning patterns, performance is further boosted to 84.3%.
>
> The 2.9% absolute accuracy gain from 81.4% (Random) to 84.3% (Ours) represents a significant empirical gain. This strongly demonstrates that the "Structured Scanning Order" is an essential component of our method, enabling xLSTM to capture complex spatial dependencies in 2D images in the way it does best (multi-directional sequence modeling).
>
> We hope these clarifications address your main concerns. The core contribution of our work is the design of the first effective, decoupled autoregressive decoding pretraining framework (MAL), which successfully unlocks the powerful performance of xLSTM in vision tasks through an innovative cluster-masked strategy and empirically validated multi-directional scanning methods.
>
> We have further strengthened these points in the final version. We sincerely hope that based on these clarifications, you might re-evaluate the contribution of our work and consider raising your score.
>
> Thank you again for your valuable time and insightful comments.

---

### Author Response · Authors · 2025-12-02
**Summary of Rebuttals: New Experiments (Table 13) & Key Clarifications**

**To the Area Chairs:**

We thank the reviewers (R1, R2, R3) for their constructive feedback and acknowledgment of our **"strong empirical results," "clear intuition," and "well-executed idea."**

Since the reviewers have not responded to our detailed rebuttals, we summarize the key clarifications and **new experiments added during the rebuttal phase** to assist your final decision. We believe we have fully addressed the concerns regarding novelty and theoretical grounding.

**1. Clarification on Novelty: Beyond "Masking" (Addressing R1 & R2)**

Reviewers questioned the novelty compared to standard MIM or ARM (Autoregressive Pretraining with Mamba). We clarified that MAL is not just a masking strategy but a **novel architectural paradigm** specifically tailored for xLSTM:

* **Decoupled Architectural Consistency:** Unlike ARM, which suffers from inconsistency (unidirectional pretraining vs. bidirectional finetuning), MAL introduces a **decoupled encoder-decoder framework**. This allows the xLSTM encoder to remain fully bidirectional (consistent with finetuning) while offloading the causal constraint to the autoregressive decoder.

* **Empirical Superiority:** We provided direct comparisons (Table 4) showing MAL outperforms standard MAE pretraining by **1.2%** and significantly surpasses previous state-of-the-art Mamba/xLSTM baselines.

**2. Justification of Scanning Orders (Addressing R1)**

R1 was concerned the scanning order was heuristic. We demonstrated that our "Alternating & All Directions" strategy is empirically foundational, not trivial.

* **Evidence:** In Table 7, our strategy yields a **2.9% accuracy gain** over random scanning (81.4% $\to$ 84.3%). This proves that for linear RNNs processing 2D data, our structured scanning is a critical component for capturing spatial dependencies, not a minor heuristic.

**3. Intuition & Architectural Necessity (Addressing R3)**

R3 asked about the "semantic meaning" and the necessity of the 2-stage design.

* **Intuition:** We clarified that predicting a "Cluster" (e.g., 4x4 patches) transforms the task from low-level texture completion to **high-level structural reconstruction**, forcing the model to learn semantic features.

* **New Ablation (Table 13, included in the revised Appendix):** During the rebuttal, we added an ablation study comparing decoder architectures. We proved that a lightweight Transformer decoder outperforms symmetric xLSTM/Mamba decoders (Top-1: 84.3% vs. 84.1%/84.0%), validating our asymmetric design principle (avoiding feature degradation in the encoder).

**4. Terminology (Addressing R2)**

We have revised the manuscript (Appendix B.1) to explicitly define "Cluster" as "Deterministic Grouped Patches" to avoid any confusion with unsupervised clustering algorithms.

**Conclusion**

MAL represents the **first successful validation of xLSTM for autoregressive visual pretraining**. By synergizing a **novel decoupled encoder-decoder architecture** with our **Innovative Cluster-Masked Masking Strategy** and **Cluster-Masked Scanning Method**, MAL effectively resolves the architectural inconsistency of prior Linear RNN works. This approach not only unlocks the scaling potential of xLSTM but also sets new benchmarks on ImageNet-1K, detection, and segmentation tasks. We hope the AC considers these clarifications and the strong performance of MAL in the final decision.

Best regards,

Authors of Submission #16949

---

### Note · Authors · 2026-01-29

I have read and agree with the venue's withdrawal policy on behalf of myself and my co-authors.

---

### Meta-Review · Area_Chair_smTj · 2025-12-19

**Summary:**

The primary concerns raised by the reviewers centered on the perceived limited novelty of the cluster-masked strategy relative to standard MAE or ARM frameworks, with specific critiques that the "cluster" terminology was misleading and the scanning order appeared heuristic rather than theoretically grounded. Reviewers also questioned the necessity of the asymmetric encoder-decoder design. While the authors effectively addressed the technical validity of these points through new ablations (Table 13) and empirical comparisons (Table 7), the theoretical proof of "optimality" for the scanning order remains limited to empirical effectiveness.

**Reviewer Concerns:**

The rebuttal effectively addressed concerns regarding novelty and design choices. The authors distinguished MAL from ARM by demonstrating the benefits of their decoupled architectural consistency, which preserves a bidirectional encoder, and validated this with a 1.2% gain over standard MAE. The critique that the scanning order was heuristic was refuted by empirical evidence showing a significant performance drop with random scanning, confirming the structural necessity of the proposed strategy. Furthermore, the authors clarified the "Cluster" terminology and provided ablations confirming that a lightweight Transformer decoder outperforms symmetric xLSTM decoders.

However, the request for a theoretical grounding of the scanning order remains partially outstanding, as the authors conceded to renaming their strategy "empirically effective" rather than providing a mathematical proof of optimality. Additionally, while the specific application to xLSTM is effective, the fundamental criticism regarding conceptual novelty persists. The method adapts known masking and grouping techniques rather than introducing entirely new learning objectives, which may still be viewed as incremental by reviewers seeking a radical departure from established paradigms.

**Reviewer Scores:**

Reviewer Q38j:  This reviewer rated the paper as "marginally below threshold" primarily due to the perception that the scanning order was heuristic and the novelty was limited. The authors provided Table 7, demonstrating a significant performance gap between random and structured scanning, effectively disproving the "heuristic" claim.

Reviewer pWEU: This reviewer’s main concerns were "misleading terminology" regarding clustering and limited novelty compared to ARM. The authors resolved the terminology issue in the Appendix and articulated the "Decoupled Architectural Consistency" as a key differentiator from ARM.

Reviewer Cir2: This reviewer was already positive but requested specific ablations regarding the decoder architecture and better intuition for the performance gains. The authors provided the exact requested ablation (Table 13), confirming the necessity of the asymmetric design, and offered a detailed explanation of "Task Difficulty" vs. "Texture Completion."

---

### Decision · Program_Chairs · 2026-01-26

Reject